# The Expression of Cannabinoid and Cannabinoid-Related Receptors on the Gustatory Cells of the Piglet Tongue

**DOI:** 10.3390/molecules29194613

**Published:** 2024-09-28

**Authors:** Rodrigo Zamith Cunha, Ester Grilli, Andrea Piva, Cecilia Delprete, Cecilia Franciosi, Marco Caprini, Roberto Chiocchetti

**Affiliations:** 1Department of Veterinary Medical Sciences, University of Bologna, 40126 Bologna, Italy; rodrigozamithcunha@gmail.com (R.Z.C.); ester.grilli@unibo.it (E.G.); andrea.piva@unibo.it (A.P.); 2R&D Division, Vetagro S.p.A., Via Porro 2, 42124 Reggio Emilia, Italy; 3R&D Division, Vetagro, Inc., 17 East Monroe Street, Suite #179, Chicago, IL 60603, USA; 4Laboratory of Cellular Physiology, Department of Pharmacy and Biotechnology (FaBiT), University of Bologna, 40126 Bologna, Italy; cecilia.delprete2@unibo.it (C.D.); cecilia.franciosi2@studio.unibo.it (C.F.); m.caprini@unibo.it (M.C.)

**Keywords:** CBR1, CBR2, GRP55, papillae foliatae, papillae vallatae, taste buds, TRPV1, TRPA1

## Abstract

The gustatory system is responsible for detecting and evaluating the palatability of the various chemicals present in food and beverages. Taste bud cells, located primarily on the tongue, communicate with the gustatory sensory neurons by means of neurochemical signals, transmitting taste information to the brain. It has also been found that the endocannabinoid system (ECS) may modulate food intake and palatability, and that taste bud cells express cannabinoid receptors. The purpose of this study was to investigate the expression of cannabinoid and cannabinoid-related receptors in the gustatory cells of the papillae vallatae and foliatae of ten piglets. Specific antibodies against the cannabinoid receptors (CB1R and CB2R), G protein-coupled receptor 55 (GPR55), transient receptor potential vanilloid 1 (TRPV1) and ankyrin 1 (TRPA1) were applied on cryosections of lingual tissue; the lingual tissue was also processed using Western blot analysis. Cannabinoid and cannabinoid-related receptors were found to be expressed in the taste bud cells and the surrounding epithelial cells. The extra-papillary epithelium also showed strong immunolabeling for these receptors. The results showed that these receptors were present in both the taste bud cells and the extra-gustatory epithelial cells, indicating their potential role in taste perception and chemesthesis. These findings contributed to understanding the complex interactions between cannabinoids and the gustatory system, highlighting the role of the ECS within taste perception and its potential use in animal production in order to enhance food intake.

## 1. Introduction

The sense of taste is essential for life; it tells us which prospective foods are nutritious while warning us of those which are toxic. The gustatory system has the function of detecting, identifying, and establishing the palatability of specific chemicals present in food and beverages [1,2]. Sugars, salts, acids, alkaloids, and amino acids can dissolve in saliva, bind to specific receptors, and activate the taste receptor cells located in the taste buds. When stimulated, those receptors activate nerve afferents which project into the brainstem, the information then propagates inside the central nervous system (CNS). Memory, hunger, satiety, and visceral changes can directly affect and can be affected by the experience of tasting [3,4]. Taste buds are the peripheral organs of gustation and are mainly located in the tongue epithelium; however, taste receptors and downstream signaling molecules are also present elsewhere in the oral cavity and digestive organs [5]. The molecular recognition of tasting, which occurs at the apical tips of the taste bud cells, ultimately results in sensory perceptions (sweet, salty, sour, bitter, umami) which guide appetite and trigger the physiological processes for absorbing nutrients and adjusting the metabolism [6]. Taste buds are multicellular organs of roughly 50–100 fusiform cells; they transduce gustatory stimuli into electrochemical signals [3,7]. Taste bud sensory cells communicate with the afferent nerve fibers of the gustatory sensory neurons via neurochemical signals, with cell bodies located in the sensory ganglia of three cranial nerves. Taste bud cells can be classified into 4 types: type I cells which are believed to function as support cells [8,9,10]; type II receptor cells which detect sweet, bitter, and umami (glutamate taste) [11,12,13]; type III cells which detect sour and salt; and type IV (or basal) cells which are generally considered to be postmitotic precursors of types I–III [14]. Type I cells are assumed to be the most numerous, representing approximately 50% of cells per bud, while type II and III cells are less common, contributing 15–20% each [15].

The sense of irritation, pungency, warmth, or cooling elicited by a wide variety of chemical compounds acting on the mucus membranes and the skin is called chemesthesis [16]. Formally, chemesthesis represents the general chemical sensitivity of the mucus membranes and skin throughout the body; this term commonly refers to sensations in the oronasal cavities and eyes (cornea).

Given this broad range of sensations, it is not surprising that the receptor mechanisms subserving chemesthesis are equally diverse and are present in different components, including sensory nociceptors, other free nerve endings, and keratinocytes; oronasal chemesthetic signals are believed to be conducted by somatosensory fibers in the trigeminal (V), glossopharyngeal (IX), and vagus (X) nerves [16]. In addition, it has been shown that, within taste buds, mutually interacting neuronal pathways may co-exist [17], indicating that other cellular signaling cascades may also play a role regarding taste perception (e.g., temperature—which can modify the perception of a taste, warm vs. cool, for instance).

Taste cells may release numerous neurotransmitters (e.g., serotonin, adenosine triphosphate [ATP], glutamate, acetylcholine, gamma-aminobutyric acid [GABA]) and express neurotransmitter receptors (serotonin, ATP, and GABA receptors), suggesting that there is communication among cells in the taste buds which may shape the output of the bud [18,19]. Several reports have indicated that the endocannabinoid system (ECS) may modulate the intake of food and its palatability and that the oral taste bud cells express cannabinoid receptors [7,20,21,22]. Various studies have demonstrated that sweet taste responses can be modulated by leptin and endocannabinoids (anandamide [AEA] and 2-arachidonoyl glycerol [2-AG]) [1,19]. Moreover, there is scientific evidence of the expression of cannabinoid-related receptors (transient receptor potential (TRP) channels, G-protein coupled receptors (GPRs), peroxisome proliferator-activated receptors (PPARs), etc.) in the taste buds [23,24]; however, in relation to food reward and hedonic values, the significance of these ‘non-canonical’ endocannabinoid receptors and ligands of the ECS remains to be elucidated [21]. Nevertheless, it is known that the trigeminal chemesthetic sensation from the mouth also derives from the activation of TRPs expressed by the free endings, which do not innervate taste receptor cells [25]. The aim of the present study was to molecularly investigate the expression of cannabinoid receptors type 1 (CB1R) and type 2 (CB2R), and cannabinoid-related receptors G protein-coupled receptor 55 (GPR55), transient receptor potential vanilloid 1 (TRPV1) and ankyrin 1 (TRPA1) in the gustatory cells of the papillae vallatae and foliatae of the pig using Western blot analysis and immunohistochemistry for the first time. Knowing the specific distribution of these receptors may create the anatomical support for additional studies to investigate new nutritional ligands and promote new anti-stress food (comfort or “comfy” food) in animal production.

## 2. Results

### 2.1. Western Blot Analysis

Western blot (Wb) analysis was carried out to determine whether papillae vallatae (or circumvallatae) and foliatae of the pig expressed proteins for CB1R, CB2R, GPR55, TRPV1, and TRPA1. Negative controls, in which the primary antibodies were not involved in the incubation with the membrane, did not show bands in any of the western blots analyzed (Figure 1A–F). furthermore, dog synovial membrane (SM) and mouse nervous system (brain and dorsal root ganglion) were used as positive controls.

The anti-CB1R antibody revealed a specific band of 70 kDa in all the pig papillae and in the positive control sample (dog SM; [26]) (Figure 1A).

The mouse anti-CB2R antibody revealed a specific band of 55 kDa in all the pig papillae and control samples (total mouse brain extract) (Figure 1B). The rabbit anti-CB2R also showed a signal in the vicinity of 30 kDa in addition to the 55 kDa band, (Figure 1C).

The anti-GPR55 antibody recognized a major band in the vicinity of 35 kDa and its dimer at 70 kDa, as well as for the positive control (dog synovial membrane) (Figure 1D).

The anti-TRPA1 antibody recognized a major band in the vicinity of 100 kDa which was present in all the pig papilla samples and in the positive controls, such as the mouse dorsal root ganglion (DRG) neurons (Figure 1E).

The anti-TRPV1 antibody recognized a major band in the vicinity of 90 kDa which was present in all the pig papillae samples [27] (Figure 1F).

### 2.2. Immunofluorescence

The cannabinoid and cannabinoid-related receptors were expressed by the taste bud gustatory and non-gustatory epithelial cells in both the papillae, with, however, differences within the intracellular distribution of the receptors and degree of immunolabeling. In addition, some receptors were also expressed by intralingual neurons (CB1R, CB2R, TRPA1, TRPV1), intralingual salivary glands (CB1R), and blood vessels (CB2R, GPR55, and TRPV1); however, only the immunostaining of the epithelial (gustatory and extra-gustatory) cells and intralingual neurons will be analyzed and discussed. As has already been described in the pig [28], the taste buds of the papillae vallatae were absent in the dorsal surface of the papillae, while they were particularly abundant in the deeper parts of the papillary sulcus. The elongated shape of the taste gustatory cells, reminiscent of that of the cloves of a garlic bulb, made the taste buds easily recognizable in the sections. In some taste buds, the nuclei were particularly clustered in the basal or central portion of the bud; this nuclear organization consequently created clearly recognizable areas with a typical enucleated aspect.

Approximately 25 dapi-labeled nuclei could be counted in single sections of a single taste bud of the vallate papillae; these data are compatible with those reported [6], having a total number (50–100 cells) of the gustatory cells of a taste bud. Approximately 10–15 taste buds could be counted in a single section of a papilla vallata. In the papillae foliatae, which were endowed with numerous crypts and epithelial lamellae in the sections, the taste buds were more numerous, and it was difficult to count them. In both the papillae, some gustatory cells (likely type I cells) showed immunoreactivity (IR) for the glial marker GFAP (Figure 2A,B) and S100 (Figure 2C,D).

A dense network of substance P (SP) immunoreactive nerve fibers and varicosities was observed in the connective tissue underlying the taste buds; SP-positive fibers were seen in proximity to unidentified cells of the taste buds. Substance P immunoreactive nerve fibers were also seen in the epithelium outside the taste buds (Appendix A). It was not unusual to observe some SP immunoreactive nerve fibers reaching the most superficial layers of the non-gustatory epithelium.

The gustatory cells were reached by a dense network of neuronal varicosities showing bright synaptophysin-IR (Appendix A).

A dense network of CB1R immunoreactive nerve fibers was visible, emerging from the tela submucosa and spreading toward the taste buds and the non-gustatory epithelial cells.

### 2.3. Cannabinoid and Cannabinoid-Related Receptors in Gustatory Cells

All the receptors studied were observed in taste bud cells, the TRPA1 being the most represented receptor, followed by TRPV1, GPR55, CB1R, and CB2R.

CB1 receptor—Generally, weak-to-moderate cytoplasmic CB1R-IR was expressed by some gustatory cells of the papillae vallatae; however, in some taste bud sections, it was not unusual to observe one or two gustatory cells expressing bright CB1R-IR (Figure 3A–C). Weak CB1R-IR was also expressed by the nuclei of the gustatory cells. Not all the taste buds showed CB1R-IR and those taste buds located more deeply in the papillae expressed more CB1R positivity. In the papillae foliatae, very weak or negative CB1R immunolabeling was observed.

CB2 receptor—Both the anti-CB2R antibodies used in the current study produced CB2R labeling of taste bud gustatory and non-gustatory cells in the papillae vallatae and foliatae. In particular, the rabbit anti-CB2R showed a granular pattern of immunolabeling, which was visible in some cells of the taste buds (Figure 3D–F).

GPR55—Some gustatory cells showed weak-to-moderate cytoplasmic GPR55-IR (Figure 4A–C). Not all taste buds showed cells immunoreactive for GPR55. The gustatory cells were more strongly labeled for GPR55 at the papillae foliatae. 

TRPV1—Taste bud cells expressed moderate-to-bright cytoplasmic TRPV1-IR which was often more concentrated in the apical part of the gustatory cells (Figure 4D–F).

TRPA1—Numerous cells of the taste buds expressed bright cytoplasmic TRPA1-IR which was evident along the entire central axis of the cells with an enhancing gradient approaching the gustatory pore (Figure 4G–I).

### 2.4. Cannabinoid and Cannabinoid-Related Receptors in Extra-Gustatory Epithelial Cells

The peri-gemmal epithelium, i.e., the epithelium surrounding the taste buds, and the extra-papillary epithelium showed intense labeling for the receptors studied.

CB1 receptor—Cannabinoid receptor 1 tended to be poorly represented in the epithelial cells; however, in all the subjects analyzed, strong CB1R-IR was detected in the epithelial cells at the bottom of the trench area below the papillae vallatae and adjacent to the opening of the ducts of the Von Ebner’s glands (VEGs). In the papillae foliatae, the CB1R-IR labelling was more marked and extended to the epithelial cells of numerous areas of the papillary crypt, adjacent to the opening of the ducts of the VEGs (Figure 5).

CB2 receptor—The CB2R-IR was weakly expressed by the cytoplasm and cellular membranes of the epithelial cells, especially from the cells of the layers of the superficial epithelium. This finding was observed following the use of both anti-CB2R antibodies utilized.

The TRPV1 and TRPA1 receptors were brilliantly expressed by the cytoplasm of the epithelial cells distributed in the epithelial layers (Figure 6). The anti-TRPA1 antibody showed the strongest level of immunolabeling of the taste cells.

### 2.5. Cannabinoid and Cannabinoid-Related Receptors in Intralingual Neurons

Small ganglia composed of roundish or ovoid intralingual neurons were visible above all beneath the papillae vallatae (Appendix A). The neurons were surrounded by GFAP- and S100-positive glial cells and by numerous varicosities immunoreactive for synaptophysin. 

The neurons exhibited bright CB1R-, CB2R-, TRPV1-, and TRPA1-IR (Figure 7). Rare neurons were also GPR55 immunoreactive. The ganglionic nerve fibers were also positive for CB1R, TRPV1, and TRPA1 (Figure 7).

## 3. Discussion

### 3.1. Cannabinoid and Cannabinoid-Related Receptors in Gustatory Epithelial Cells

This is the first evidence of cannabinoid (CB1R and CB2R) and cannabinoid-related (GPR55, TRPV1, and TRPA1) receptors in the gustatory and non-gustatory epithelial cells of the pig tongue.

A recent review has demonstrated a clear link between eating behavior and the ECS [29].

There is a convincing amount of evidence showing that the ECS may influence the appetite and taste preference of rodents but at different levels. In the hypothalamus and limbic forebrain, endocannabinoids seem to be able to induce appetite and stimulate food intake by acting on the CB1R, a function that opposes the action of leptin [30]. However, the systemic administration of endocannabinoids and exogenous cannabinoids causes hyperphagia [20] and increases the preference for palatable substances, such as sucrose solution and sweetened food pellets [31].

Studies regarding animals and humans have indicated that 2-AG and AEA may stimulate hunger and food intake by interacting with CB1R [32,33]. More specifically, 2-AG or AEA selectively increased behavioral responses to sweeteners without affecting responses to salty, sour, bitter, and umami compounds [7]. Mice genetically lacking CB1R showed no such enhancement of sweet taste responses by endocannabinoids, and the sweet-enhancing effect was prevented by a CB1R antagonist, indicating that the effect could be mediated by CB1R. It was demonstrated that CB1R was responsible for increased food intake [20], induced by an endocannabinoid agonist. Cannabinoid receptor 1 may also be involved in fat taste perception; in fact, a reduced preference for fat among CB1R knockout mice has been shown [22].

In contrast to the initial findings, the distribution of the CB1R is not limited to the central nervous system (CNS), and the involvement of the CB1R in food intake regulation may occur at different levels, even within the gastrointestinal tract, to regulate hedonic reward in the brain. It is already known that the stimulation of appetite and feeding behavior is linked to reward and rewarding behavior [30]; it is associated with the release of dopamine [34] which cannabinoids can also induce [35].

In the present study, the immunoreactivity for both cannabinoid receptors have been observed in the gustatory cells. These findings are partially consistent with those reported by Yoshida et al. (2010) [7] who observed CB1R-IR in gustatory type II cells of the fungiform and vallatae papillae of mice and the mRNA of CB2R only in the extra-gustatory epithelial cells.

An increasing amount of evidence suggests an important physiological role for GPR55 [36]. However, very few investigations have correlated the expression of GPR55 with tasting and/or chemesthesis. An abundance of mRNA of the CB1R and GPR55 in the tongue epithelia of cows was shown; in early lactation, AEA promotes a preference for sweet-tasting feed [37]. It is known that maternal milk contains high levels of endocannabinoids, such as 2-AG, AEA, and other fatty acids [38]. Therefore, the ECS and cannabinoids are important in feeding control so much so that they may be involved in mammalian early-life feeding (which is crucial for survival and development) [39]. In the current study, weak-to-moderate GPR55-IR was observed in the taste cells which was more strongly expressed in the gustatory cells of the papillae foliatae.

Taste induces changes in Ca^2+^ levels, membrane potential, and pH (sour testing) in the taste cells of the tongue, and/or the neurons involved in the transmission and decoding of the taste signals to the brain [17,40,41]. As was also shown in the current study, distinct subsets of cells in the taste buds may express TRP channels in which the gustatory cells of both the papillae vallatae and foliatae showed bright TRPV1 and TRPA1 immunostaining. While TRPV1 is not expressed in the taste receptor cells of rodents [40,42], it is instead expressed in cultured human taste cells isolated from fungiform papillae [43,44].

A recent study reported that TRP cation channels were expressed by type III gustatory cells [40], the only taste cells having ultrastructural specialization associated with synapses that respond directly to acid (sour) taste stimuli. Type III cells may also respond to high concentrations of NaCl and KCl and, indirectly (via the purinergic paracrine pathway from type II receptor cells), to sweet, bitter, and umami tastes [16].

Two sensations related to gustation (salty taste and pungency) are believed to involve TRPV1, although, for the most part, spicy ingredients seem to excite somatosensory afferents (i.e., chemesthesis, see below). The pepper compound capsaicin is one of the most potent TRPV1 agonists. In addition, TRPV1 is also sensitive to vanillin, temperature (42–53 °C), and pH. It has been proposed as being responsible for salt detection since it mediates responses of the chorda tympani nerve (nervus intermedius), not only to Ca^2+^ but also to Na^+^ and NH^4+^ [44]. To confirm this last piece of evidence, it has been shown that polymorphisms of the TRPV1 gene were associated with alterations in salty taste sensitivity and salt preference [45].

Transient receptor potential vanilloid 1 and TRPA1 are often co-expressed in subpopulations of sensory neurons, keratinocytes, and inflammatory cells [46,47,48] in which the two receptors seem to form a complex at the plasma membrane and influence the functioning of each other [40]. Tastants, such as mustards, garlic, and turmeric, selectively activate TRPA1, whereas other tastants may simultaneously activate TRPA1 and TRPV1 (cloves, ginger, and black pepper) [49]. These spices and herbs contain molecules acting on chemesthesis; however, there is evidence supporting the fact that TRPA1 and TRPV1 indirectly interact with the gustatory system [40]. Additional investigation is necessary since, in the current study, the most expressed immunolabeling in the taste cells was found in TRPA1 and TRPV1. Therefore, at least in pigs, it seems that these receptors could be involved in taste signals.

### 3.2. Cannabinoid and Cannabinoid-Related Receptors in the Extra-Gustatory Epithelial Cells

The oral cavity is the primary site of ingestion and is exposed to a wide range of chemical, physical, and/or thermal stimuli as compared with other regions of the body. It is believed that oral sensation, except for taste perception, is received by nerves distributed close to the oral epithelium. Therefore, it is of paramount importance that the oral epithelium be able to detect thermal, chemical, irritant, and mechanical stimuli, and also transmit information to the neighboring nerves which produce chemesthesis [50].

For instance, the consumption of spices and herbs elicits the perception of burning, pungency, irritation, cooling, or warmth in the mouth. These sensations are generally not considered to directly interact with the gustatory system but contribute to the chemical sense referred to as chemesthesis [51]. In the current study, CB1R, CB2R, GPR55, TRPV1, and TRPA1 expression were also observed in the chemosensory extra-gustatory epithelial cells of piglets.

Many of the plant-derived molecules having chemesthetic properties activate TRP channels, such as TRPA1 and TRPV1 [52], which are expressed not only by taste buds (and nerve fibers) but also by the extra-gustatory epithelial cells of the tongue and oral mucosa [50,53].

### 3.3. Receptors in Intralingual Neurons

Nerve cell bodies are localized in the small ganglia, located for the most part, close to the muscle layers, near the salivary VEGs, and in the tunica propria, predominantly spotted at the base of the papillae vallate, known as the circumvallate ganglia [54]. Nerve fibers run in the tunica propria from the circumvallate ganglia and innervate the glands of the region [54].

The intralingual ganglia are part of the sensory apparatus of the tongue and play a role within the intralingual reflex arch [55,56].

Circumvallate ganglia are considered to be the post-ganglionic parasympathetic neurons (the preganglionic neurons belong to the salivatory nuclei of cranial nerves VII and IX) involved in salivary secretion and blood vessel dilatation [57]. In addition, the circumvallate ganglia are also considered to be the most rostral part of the enteric nervous system (oral nervous system) [54].

The expression of bright CB1R-IR at the circumvallate ganglia better describes the role of the CB1R in regulating glandular secretion [58] and vasoconstriction. These neurons could play a role in the gustatory-salivary reflex [59] and could explain the fast dry-mouth effect of Tetrahydriocannabidiol (THC), an agonist of CB1R, by the intralingual arch reflex [60].

Cannabinoid receptor 1 positive neurons at the circumvallate ganglion point to the role of the lingual plexus as the rostral part of the enteric nervous system (ENS), potentially involved in the brain-gut axis, since CB1 receptors are known to play a role in the appetite regarding fat metabolism [61,62].

The function of CB2R on the intralingual neurons is not fully understood/clear. However, there is morphological evidence that CB2R-IR is expressed by the cytoplasm of the neurons localized in the mandibulary glands of piglets [63] and pharmacological evidence showing the influence of CB2R in salivatory regulation [64,65].

The expression of TRPV1 by the cytoplasm of the circumvallate neurons may explain the fast response when exposed to thermic stimuli or ″spicy″ molecules, with high temperatures and capsaicin both being agonists of the TRPV1 [66]. Similarly, the expression of neuronal TRPA1-IR may be associated with the fast response to menthol or cinnamaldehyde, or to allicin (from garlic extract), and acroline (from diesel exhaust) which are both agonists of the TRPA1 [66].

Taken together, this evidence suggests that the endocannabinoid tone may influence palatability, and consequently stimulate the desire to eat and the hedonic value of foods [21]. Additional investigation is needed to fully understand the role of these receptors at the gustatory organ level and to elucidate the pathways of integration between the endocannabinoid system, the ENS, and the hedonic value of food.

Limitation—The subclasses of gustatory cells expressing the receptors studied were not identified; at present, no markers have been studied and produced that can identify the cellular subtypes of taste buds in this species. Older animals were not included in the present study; therefore, it is not possible, at present, to understand whether some receptors modify their expression during aging.

## 4. Material and Methods

### 4.1. Animals

Tissue was collected from 10 piglets, hybrids (Landrace; Large White; Duroc) (weight 10.9 ± 2.2 Kg having an average age of 47 days (26 lactating) which were slaughtered 21 days post-weaning (Authorization no. 287/2021-PR [Resp. To prot. 2216A.19]).

The roots of the tongues (radix linguae), with papillae vallatae and foliatae, were collected within 20 min of the animals’ deaths.

Small parts of the tongue, each containing the papillae vallatae and foliatae, were removed using a scalpel and were immediately immersed in the fixative (4% paraformaldehyde and 0.1 M sodium phosphate buffer pH 7.0) at +4 °C for 48 h. After rinsing in phosphate-buffered saline (PBS), the tissues were stored in PBS containing sucrose (30%) and sodium azide (0.1%) (pH 7.4) at +4 °C. The tissues were subsequently transferred to a mixture of PBS-sucrose-azide and OCT compound (Tissue Tek^®^, Sakura Finetek Europe, Alphen aan den Rijn, The Netherlands) in a ratio of 1:1 (12 h), and were then immersed in 100% OCT.

The tissues were placed in cryomold trays (Tissue Tek^®^) containing a 100% solution of Tissue tek (100%) and were frozen in isopentane (2-methyl-butane) cooled in liquid nitrogen. The frozen tissues were then stored at −80 °C until sectioning (thickness: 14 μm), using a cryostat. The sections were collected on poly-L-lysine coated slides and subsequently processed for immunofluorescence.

For the Wb analysis, the tissues were placed in sterilized Eppendorf tubes which were immersed in liquid nitrogen and were then stored at −80 °C.

### 4.2. Immunofluorescence

The cryosections were hydrated in PBS and processed for immunostaining. To block non-specific binding, the sections were incubated in a solution containing 20% normal donkey serum (Normal donkey serum, Colorado Serum Co., Denver, CO, USA), 0.5% Triton X-100 (Sigma Aldrich, Milan, Italy, Europe), and bovine serum albumin (1%) in PBS for 1 h at room temperature (RT). For single immunostaining, cryosections were incubated in a chamber overnight at room temperature with antibodies (obtained from rabbits) directed against the five receptors studied (CB1R, CB2R, GPR55, TRPV1, and TRPA1) (Table 1). For double immunostaining, the cryosections were incubated with mixtures of two different primary antibodies (Table 1). The primary antibodies were diluted in 1.8% NaCl in 0.01 M PBS, containing 0.1% sodium azide.

After washing the sections in PBS (3 × 10 min), they were incubated for 1 h at RT in a humid chamber with secondary antibodies (Table 2) diluted in PBS.

The cryosections were then washed in PBS (3 × 10 min) and mounted in glycerol buffered at pH 8.6 with 4, 6-diamidino-2-phenylindole-Dapi- (Santa Cruz Biotechnology, CA, USA).

#### Specificity of the Primary Antibodies

CB1R—The immunogen used to obtain the anti-CB1R antibody was the synthetic peptide MSVSTDTSAEAL, corresponding to the carboxy-terminal amino acids 461–472 of the human CB1 receptor. The homology between the pig (F1S0E6) and the human (P21554) complete amino acid sequences of the CB1 receptor was 97.9% (https://blast.ncbi.nlm.nih.gov/Blast.cgi, accessed on 7 January 2018) and the correspondence with the specific immunogen sequence was 100%. Therefore, the human anti-CB1R antibody should recognize the same receptor in pigs as well. The specificity of human tissue has recently been verified by the present research group on human wholemount preparation [26,67]. The specificity of this antibody was also tested using Wb analysis also in the current study (Figure 1A).

CB2R—Two different anti-CB2R antibodies were used.

The immunogens used to obtain the rabbit anti-CB2R recombinant monoclonal antibody (Thermo fisher, 13H43L20) were the peptides corresponding to human CB2R (1: aa341–aa360, 2: aa5–aa25, 3: aa2–25). The specificity of the rabbit anti-CB2R antibody was tested on pig tissues using Wb analysis in the current study (Figure 1B).

The immunogen used to obtain the other mouse anti-CB2R antibody (sc-293188) was human-origin CB2R amino acid sequence 302–360 (P34972). The homology between the complete amino acid sequences of the pig and the human CBR2 was 81.9%. The homology of the specific amino acid sequence (RSGEIRSSAHHCLAHWKKCVRGLGSEAKEEAPRSSVTETEADGKITPWPDSRDLDLSDC) with the porcine species was 70.69% (https://blast.ncbi.nlm.nih.gov/Blast.cgi, accessed on 7 January 2018). The same mouse anti-CB2R antibody has already been used on porcine tissues [68]. The specificity of the mouse anti-CB2R antibody was proven on pig tissues using Wb analysis in the current study (Figure 1C).

GPR55—The immunogen used to obtain the anti-GPR55 antibody was the synthetic 20 amino acid peptide from the third cytoplasmic domain of Human GPR55 in amino acids 200–250. The homology between the full amino acid sequences of the pig and human GPR55 was 80%. The specificity of the rabbit anti-GPR55 antibody has been evaluated on pig tissues with Wb analysis in the current study (Figure 1D).

TRPA1—The immunogen used to obtain the anti-TRPA1 antibody was a synthetic peptide from rat TRPA1 conjugated to blue carrier protein. The alignment of the immunogen sequence with the target protein in the pig was 80.3% (https://blast.ncbi.nlm.nih.gov/Blast.cgi, accessed on 7 January 2018). This antibody was also tested using Wb analysis in the current study (Figure 1E).

TRPV1—The immunogen used to obtain the anti-TRPV1 antibody was the (C) EDAEVFK DSMVPGEK peptide, corresponding to residues 824–838 of the rat TRPV1. The homology between the complete amino acid sequences of the pig (A0A4X1UCR0) and rat (O35433) TRPV1 was 84.52% (https://blast.ncbi.nlm.nih.gov/Blast.cgi, accessed on 7 January 2018), and the correspondence with the specific immunogen sequence was 93%. This antibody had already been tested by the present research group on the pig nervous system using Wb analysis [27]. In addition, this antibody was also tested using Wb analysis in the current study (Figure 1F).

### 4.3. Specificity of the Secondary Antibodies

The specificity of the secondary antibodies was tested by applying them to the sections after omitting the primary antibodies. No immunolabeled cells were detected after omitting the primary antibodies (Appendix A).

### 4.4. Semiquantitative Analysis of the Immunoreactivity

The immunoreactivity of the antibodies was evaluated, and their cellular localization (membranous and cytoplasmic) was reported.

### 4.5. Fluorescence Microscopy

The preparations were examined by the same observer using a Nikon Eclipse Ni microscope 227 (Nikon Instruments Europe BV, Amsterdam, The Netherlands, Europe) equipped with the appropriate filter cubes. The images were recorded using a DS-Qi1Nc digital camera and NIS Elements software BR 4.20.01 (Mountain View, Ottawa, ON, Canada). Slight contrast and brightness adjustments were made using Corel Photo Paint whereas the figure panels were prepared using Corel Draw (Mountain 232 View). 

### 4.6. Western Blot Analysis

Tissue samples (papillae vallate and foliatae) were collected from three piglets; they were frozen and stored at −80 °C until sample processing. Tissues at the amount of 50 mg were fractioned into small pieces and homogenized in 500 µL of RIPA buffer (50 mM TRIS-HCl, pH 7.4, 100 mM NaCl, 1 mM PMSF, 1 mM EDTA, 5 mM Iodoacetamide 1% Triton X-100, and 0.5% Sodium dodecysulphate) supplemented with a protease inhibitor cocktail (Sigma-Aldrich, Co, St. Louis, MO, USA). The extract was sonicated for 10 min in 20 s intervals every 2 min and pelleted for 20 min (14,000 rpm) at 4 °C. The total protein content was determined using the Bradford method. Proteins (from 7.5 to 10 µg) were separated using 10–12% SDS-polyacrylamide gel and transferred to a polyvinylidene fluoride (PVDF) membrane. After transfer, the membrane was blocked by 5% milk powder in PBS-T (PBS 0.01M, pH 7.4) with 0.05% Tween 20 (Sigma) for 1 h at RT. The membranes were incubated with primary antibodies (rabbit anti-CB1R; rabbit anti-CB2R; mouse anti-CB2R; rabbit anti-GPR55; rabbit anti-TRPV1; and rabbit anti-TRPA1) overnight at 4 °C and were diluted 1:1000 in PBS-T containing 1% milk. The following day, the membranes were washed 3 times with PBS-T, for 15 min each, and immunoglobulin G (IgG) horseradish peroxidase-conjugated secondary antibodies mouse anti-rabbit (1:75,000, Santa Cruz) was utilized for incubation in 1% milk powder in PBS-T for 1 h at RT. After 3X washings of the secondary-horseradish peroxidase (HRP) binding antibody, membrane was incubated with chemiluminescence substrate and developed with the enhancing chemiluminescence detection system (Santa Cruz Biotechnology or Cyanagen–Westar ηC ultra 2.0). The blots were visualized using the ChemiDocTM (Bio-Rad, Hercules, CA, USA) imaging system.

## 5. Conclusions

The diversity of cannabinoid receptors localized, not only in taste cells but also in the intralingual neurons and epithelial cells, points to a potentially key role played by the ECS within taste coding and may exert a major function regarding the hedonic value of food and its rewarding behavior. These results better described the role of the complex gustatory system and the axis between the oral cavity and the ENS. The expression of ECS receptors at the taste cells and intralingual neurons points to a potential role on the neuro-enteric-behavioral axis, creating such a complex cellular signaling network/framework that not only perceives taste but is also able to decode a variety of stimuli from the oral cavity and transmit the response to the CNS and the ENS.

## Figures and Tables

**Figure 1 molecules-29-04613-f001:**
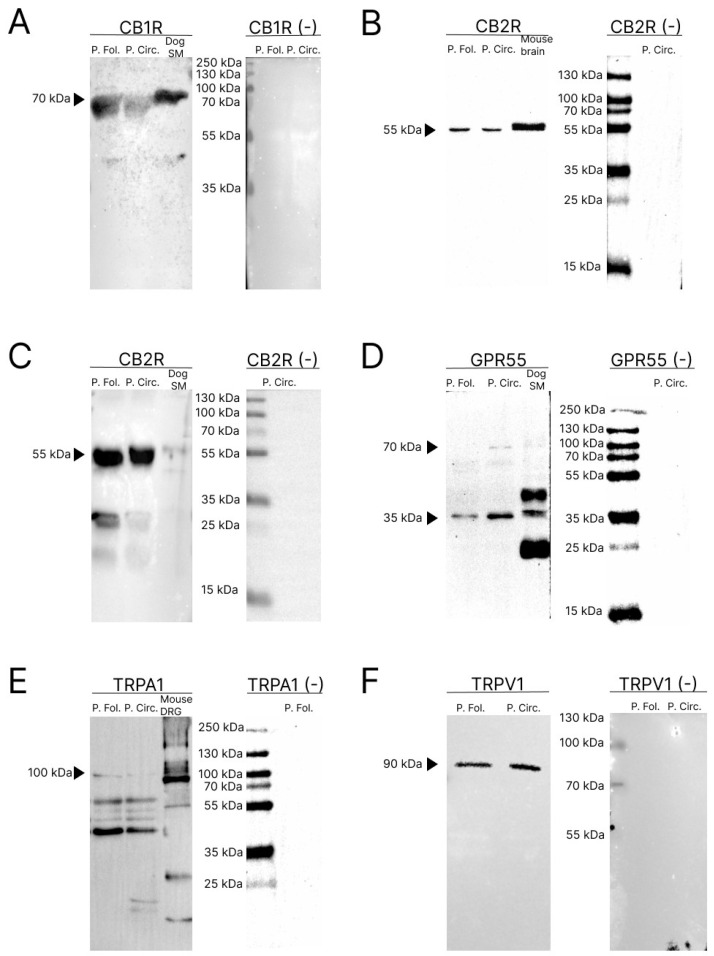
Representative image of Western blot analysis showing the specificity of the primary antibodies utilized: rabbit anti-cannabinoid receptor 1 (**A**), mouse anti-cannabinoid receptor 2 (**B**), rabbit anti-cannabinoid receptor 2 (**C**), rabbit anti-G protein-coupled receptor 55 (GPR55) (**D**), rabbit anti transient receptor ankyrin 1 (TRPA1) (**E**), and rabbit anti transient receptor vanilloid 1 (TRPV1) (**F**). Negative controls, in which the primary antibodies were not involved in the incubation with the membrane, did not show bands (right panels). Abbreviations: DRG, dorsal root ganglion; P. Circ., circumvallatae papillae; P. Fol., foliatae papillae.

**Figure 2 molecules-29-04613-f002:**
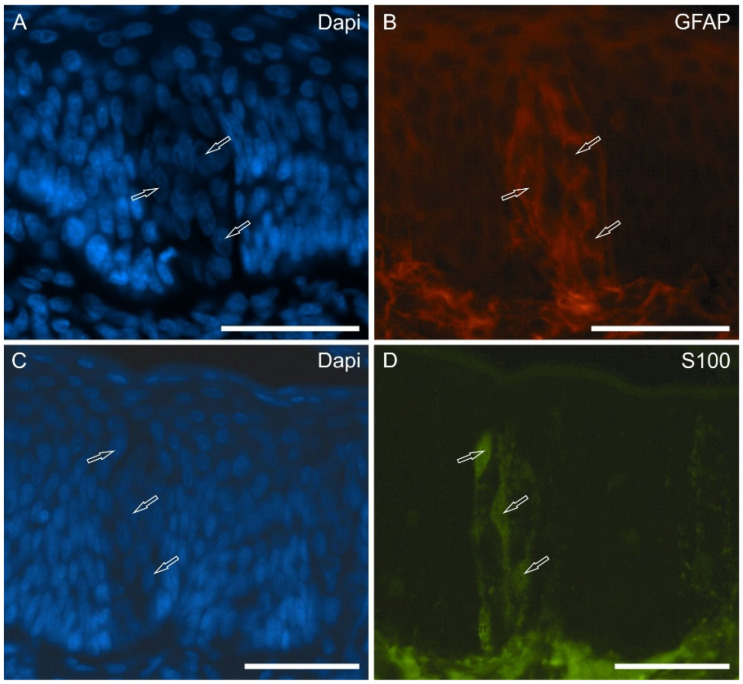
Photomicrographs showing cryosections of the piglet papillae vallatae in which the anti-GFAP and -protein S100 antibodies were applied. The arrows indicate some dapi-labeled nuclei (**A**,**C**) of cells belonging to the taste buds which showed faint-to-moderate immunoreactivity for the glial marker GFAP (**B**) and S100 (**D**). Scale bar: 50 µm.

**Figure 3 molecules-29-04613-f003:**
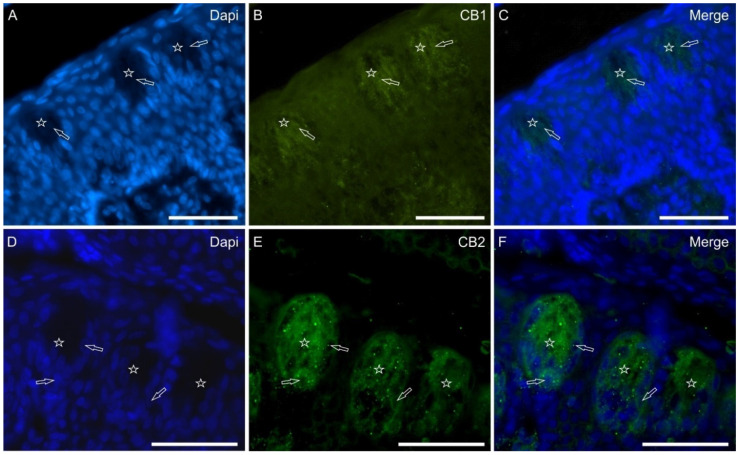
Photomicrographs showing cryosections of the piglet papilla vallata (**A**–**C**) and papilla foliata (**D**–**F**) in which antibodies against cannabinoid receptors type 1 (CB1) and type 2 (CB2) were used. The stars indicate contiguous taste buds; the empty arrows indicate the nuclei of taste bud cells which were positive for CB1 (**B**) and CB2 (**E**) receptors immunoreactivity. (**C**,**F**): merged images. Scale bar: 50 µm.

**Figure 4 molecules-29-04613-f004:**
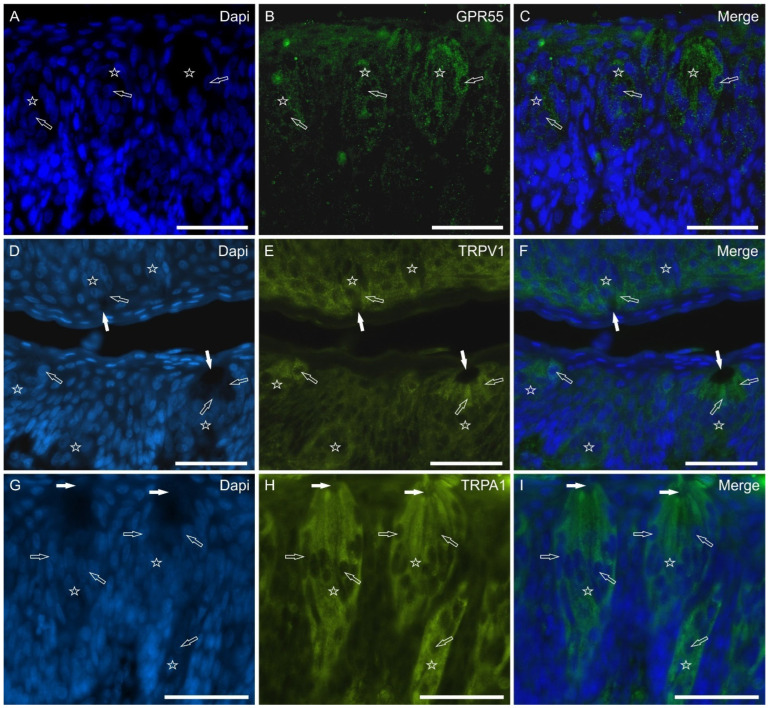
Photomicrographs showing cryosections of the piglet papillae vallatae (**A**–**C**,**G**–**I**) and foliatae (**D**–**F**) in which anti-GPR55, TRPV1, and TRPA1 antibodies were applied. The stars indicate the taste buds. The white arrows indicate the taste pores of some taste buds; the empty arrows indicate the nuclei of taste cells positive for GPR55 (**A**–**C**), TRPV1 (**D**–**F**), and TRPA1 (**G**–**I**). (**C**,**F**,**I**): merged images. Scale bar: 50 µm.

**Figure 5 molecules-29-04613-f005:**
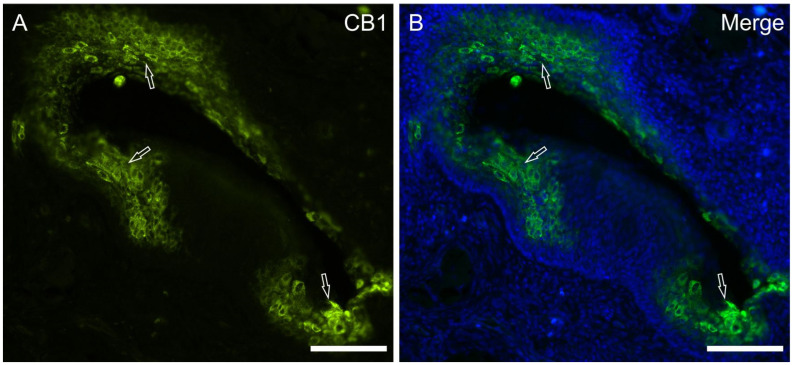
Photomicrographs showing a cryosection of the piglet papilla foliata in which anti-CB1 receptor antibody was applied (**A**). CB1 receptor immunoreactivity was observed, in all the papillae foliatae analyzed, in the clusters of the epithelial cells located in proximity to the crypts (arrows). (**B**): merged image. Scale bar: 50 µm.

**Figure 6 molecules-29-04613-f006:**
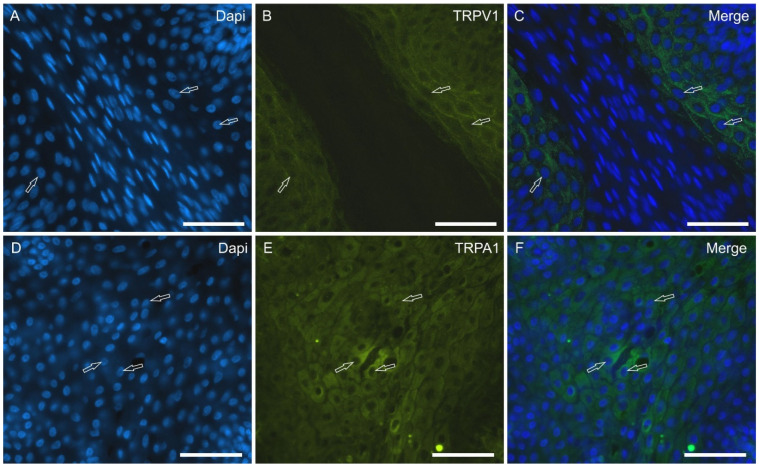
Photomicrographs showing tangential cryosections of the surface of the piglet tongue in which anti-TRPV1 and TRPA1 antibodies were applied. The arrows indicate the nuclei of some of the most superficial non-gustatory epithelial cells showing bright TRPV1 (for the most part on the cell membrane) (**A**–**C**) and TRPA1 (for the most part within the cytoplasm) (**D**–**F**) immunoreactivity (**B**); (**C**,**F**): merged images. Scale bar: 50 µm.

**Figure 7 molecules-29-04613-f007:**
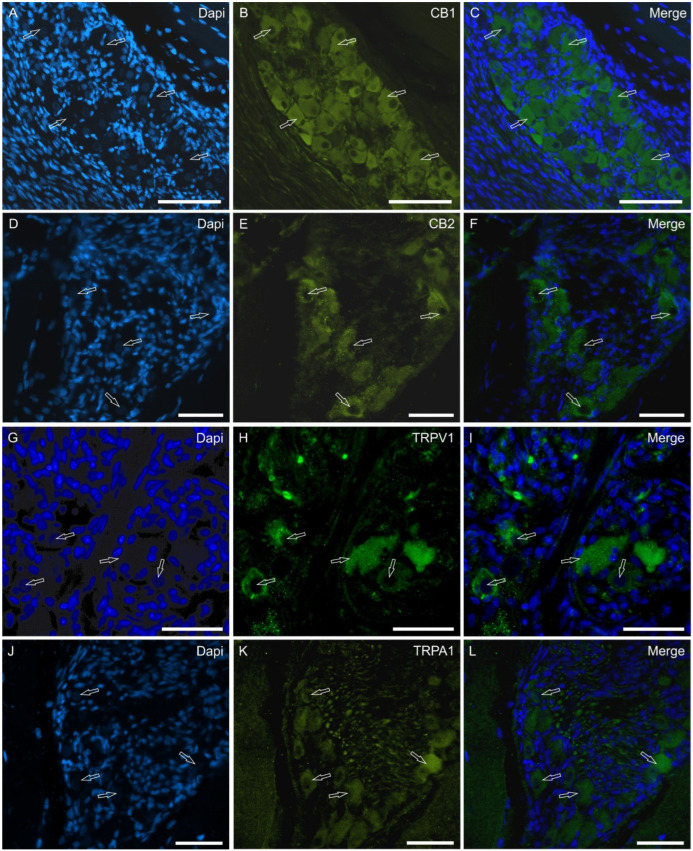
Photomicrographs showing cryosections of the piglet papillae vallatae in which large ganglia of lingual neurons exhibited bright CB1R (**A**–**C**), CB2R (**D**–**F**), TRPV1 (**G**–**I**) and TRPA1 (**J**–**L**) immunoreactivity (arrows). (**C**,**F**,**I**,**L**): merged images. Scale bar: 50 µm.

**Table 1 molecules-29-04613-t001:** Primary antibodies utilized in the study.

Primary Antibody	Host	Code	Dilution	Source
CB1R	Rabbit	ab23703	1:100	Abcam
CB2R	Mouse	sc-293188	1:50	Santa Cruz
CB2R	Rabbit	13H43L20	1:250	Thermo Fisher
GFAP	Chicken	ab-4674	1:800	Abcam
GPR55	Rabbit	NB110-55498	1:100	Novus Biol.
S100	Rabbit	PC-157	1:50	Oncogene
SP	Rat	10-515A	1:500	Fitzgerald
Synaptophysin	Rabbit	ab14692	1:100	Abcam
TRPA1	Rabbit	100-91319	1:400	Novus Biol.
TRPV1	Rabbit	ACC-030	1:200	Alomone

Suppliers of the primary antibodies: Abcam, Cambridge, UK; Alomone, Jerusalem, Israel; Fitzgerald Industries Int., Inc., Concord, MA, USA; Novus Biologicals, Littleton, CO, USA; Oncogene Research Products, La Jolla, CA, USA; Santa Cruz Biotechnology, Paso Robles, CA, USA; Thermo Fisher Scientific, Waltham, MA, USA.

**Table 2 molecules-29-04613-t002:** Secondary antibodies utilized in the study.

Secondary Antibody	Host	Code	Dilution	Source
Anti-mouse IgG Alexa-594	Donkey	A-21203	1:500	Thermo Fisher
Anti-rat 594	Donkey	A-21209	1:500	Thermo Fisher
Anti-rabbit 488	Donkey	A-21206	1:1000	Thermo Fisher
Anti-chicken TRITC	Donkey	703-025-155	1:200	Jackson

Suppliers of the secondary antibodies: Jackson ImmunoResearch Laboratories, West Grove, PA, USA; Thermo Fisher Scientific, Waltham, MA, USA.

## Data Availability

Data are contained within the article and Appendix A.

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
