# Peer review of "The Expression of Cannabinoid and Cannabinoid-Related Receptors on the Gustatory Cells of the Piglet Tongue"

_molecules, 2024, doi:10.3390/molecules29194613_

Round 1

Reviewer 1 Report

Comments and Suggestions for Authors

Cunha et al reports on the expression of receptors for the endocannabinoids and Trp channels in porcine taste papillae and the tongue in general. The primary methods employed are immunohistochemistry and western blot. They show the expression of these receptors in taste papillae. I have several reservations about the immunohistochemistry data provided. No colocalization with taste cell subtypes using taste marker proteins (sweet, type II etc.) have not been done. Thus, it is not possible to determine if the ECS in pig taste cells is similar to that in mice (to authors do mention this in the limitations). Second, there are no control experiments done to show that the antibody staining they observe is real. Indeed, many of the antibodies show weak but extensive staining, as would be expected for background. The best in this model would be if preincubation of antibody with the immunogenic peptide would block the staining in the tissue. Omitting primary antibody is a poor substitute (the result for these experiments is quite appropriate to show as a supplementary figure). The authors claim a receptor is located to the membrane/ cytoplasm in several cases (e.g., Figure 6), but in the absence of costaining with a membrane protein or higher resolutions images, these claims are not valid.  

Additional minor comments are below:

Line 44- taste buds are not present on digestive organs- rather taste receptors and down stream signaling molecules are expressed in these organs.

Line 56-7; type IV cells are not stem/progenitor cells. They are postmitotic precursors of types I-III cells.

reference 18 has nothing to do with chemesthesis.

Figure 1- please expand abbreviations in the figure.

Figure 2- why is this figure important?

Author Response

Cunha et al reports on the expression of receptors for the endocannabinoids and Trp channels in porcine taste papillae and the tongue in general. The primary methods employed are immunohistochemistry and western blot. They show the expression of these receptors in taste papillae.

I have several reservations about the immunohistochemistry data provided. No colocalization with taste cell subtypes using taste marker proteins (sweet, type II etc.) have not been done. Thus, it is not possible to determine if the ECS in pig taste cells is similar to that in mice (to authors do mention this in the limitations).

Response: As the Reviewer pointed out, we have acknowledged this limitation in the manuscript. Research on large mammals poses significant challenges, particularly when using antibodies which are specifically designed for human and rodent tissues. While extensive investigations have been carried out on these smaller species, such findings cannot always be directly applied to pigs. Although the gustatory mechanisms in pigs may share similarities with rodents, the distinct types of taste cells in pigs have yet to be identified.

Our study represents the first investigation of the endocannabinoid system (ECS) in pig taste buds, despite the aforementioned limitations. The primary objective was to demonstrate the presence of ECS receptors in taste buds, and we hope that this research will encourage additional studies in pigs, potentially focusing on identifying and characterizing the phenotypes of the different taste cell subtypes.

We have added a sentence to the ‘Limitations’ section to clarify that, to date, no markers are available to differentiate the cellular subtypes of taste buds in this species.

Second, there are no control experiments done to show that the antibody staining they observe is real. Indeed, many of the antibodies show weak but extensive staining, as would be expected for background. The best in this model would be if preincubation of antibody with the immunogenic peptide would block the staining in the tissue.

Response: The gold standard for validating antibody specificity is to stain tissues with and without the antigen of interest. This is typically done by comparing tissue from a wild-type animal with one in which the antigen has been eliminated via transgenic engineering. However, the critical question here is: are transgenic pig models available for all the receptors we studied? The answer, unfortunately, is no.

In the absence of knockout models, a more feasible, but still effective approach, can be utilized. First, Western blot analysis should be carried out on the tissue of interest to confirm that the antibody recognizes only one specific antigen and that it corresponds to the appropriate molecular weight.

As an additional specificity test, adsorption controls should be used, as suggested by the Reviewer. However, these tests must be carried out with caution. While adsorption controls are often considered to verify antibody specificity, they are not always sufficient in every context. It is also important to note that many companies do not provide the specific peptide needed for adsorption tests.

Without creating undue complexity, we applied the most specific test available under these circumstances—Western blot (Wb)—to confirm the specificity of the antibodies in the absence of knockout pigs.

We generally adhere to the guidelines provided by Clifford B. Saper and Paul E. Sawchenko in their article 'Magic Peptides, Magic Antibodies: Guidelines for Appropriate Controls for Immunohistochemistry' (The Journal of Comparative Neurology, 465:161–163, 2003).

Omitting primary antibody is a poor substitute (the result for these experiments is quite appropriate to show as a supplementary figure).

Response: A new supplementary figure (supplementary Fig. S4) has been prepared. The figures show two adjacent sections of the vallata papilla; in one section, the anti-TRPV1 antibody was applied whereas, in the adjacent section, only the secondary antibody was utilized (negative control). We think that showing the negative control of only one receptor (in this case TRPV1) was enough because we utilized the secondary antibody donkey anti-Rabbit 488 to detect all the cannabinoid and cannabinoid-related receptors which, as the Reviewer can see, did not produce any aspecific immunolabeling.

The authors claim a receptor is located to the membrane/ cytoplasm in several cases (e.g., Figure 6), but in the absence of costaining with a membrane protein or higher resolutions images, these claims are not valid. 

Response:  In response to this concern, I would like to point out that I have extensive experience using both confocal and fluorescence microscopy, having published many papers utilizing these techniques. While confocal microscopy is often highly regarded, there may be an overestimation of its advantages as compared to the fluorescence microscopy which we used in this study.

When the tissue section is of high quality, the antibody is specific, and the observer is highly experienced, it is possible to accurately differentiate between membrane and cytoplasmic labeling. This observation would be more relevant if we were describing intricate structures, such as vescicles with different contents, mitochondria, ribosomes, or lysosomes, in which the resolution might be critical. However, in this case, as we have already demonstrated in several of our previous publications, there is little doubt regarding the identification of the labeling pattern.

We believe that the combination of our experience, the specificity of the antibodies, and the high-quality sections used provides sufficient confidence in our claim regarding the receptor localization.

Lists of publications:
https://pubmed.ncbi.nlm.nih.gov/12462358/
https://pubmed.ncbi.nlm.nih.gov/12483280/
https://pubmed.ncbi.nlm.nih.gov/15338268/
https://pubmed.ncbi.nlm.nih.gov/15912404/
https://dx.doi.org/10.3390/IJMS242115949
https://dx.doi.org/10.3390/ANI13182833
https://dx.doi.org/10.3389/FVETS.2023.1045030

Additional minor comments are below:

 Line 44- taste buds are not present on digestive organs- rather taste receptors and down stream signaling molecules are expressed in these organs.

Response: Taste buds have also been described in the esophagus of human fetuses (Ponzo, 1907) and adults (Schinkele, 1942; Burkl, 1954) (In: Martin Witt, 2019: Anatomy and development of the human taste system. Handbook of Clinical Neurology, Vol. 164 - 3rd series). This reference has been added and the sentence has been changed accordingly using the input of the Reviewer.  We changed the reference [5] (and then inserted both references): Taste receptor signaling in the mammalian gut E Rozengurt, C Sternini - Current opinion in pharmacology, Volume 7, Issue 6. December 2007, Pages 557-562) + Ponzo, 1907) and adults (Schinkele, 1942; Burkl, 1954).

Line 56-7; type IV cells are not stem/progenitor cells. They are postmitotic precursors of types I-III cells.

Response: We thank the Reviewer for this precise clarification. The sentence has been changed accordingly.

reference 18 has nothing to do with chemesthesis.

Response: The reference has been removed in the sentence describing “chemesthesis”.

Figure 1- please expand abbreviations in the figure.

Response: The abbreviations have been expanded, accordingly.

Figure 2- why is this figure important?

Response: Type I cells have functions similar to those of glial cells in the central nervous system. Therefore, immunolabeling with antibodies against GFAP and S100 (glial markers) suggests that, in the pig taste buds, some cells resembling glial cells (putative type I cells) are present and can be identified. In addition, it has been proposed that a subpopulation of type I cells may be involved in salt taste reception (Vandenbeuch et al., 2008. Amiloride-sensitive channels in type I fungiform taste cells in mice. BMC Neurosci 9: 1). However, it remains unknown whether these cells serve the same function in pigs.

Reviewer 2 Report

Comments and Suggestions for Authors

The authors aimed to investigate the expression of cannabinoid receptors and cannabinoid-related receptors in taste buds and other lingual tissues. The results indicate that the immunosignals of these receptors were expressed in the taste bud cells and the surrounding epithelial cells. The extra-papillary epithelium also showed strong immunolabeling for these receptors. The data suggests potential roles of cannabinoid system in taste perception. The writing of the article needs to be improved significantly.

1.      Abstract: Conclusions in line 24-26 are not clear regarding why the “presence of CB-receptors in the extra-gustatory epithelial cells” indicate “their potential role in chemesthesis. Is it the somatosensory nerve fibers or the “extra-gustatory epithelial cells” that transduce the chemesthesis signals?

2.      Introduction:

Overall, it could be improved by being more concise and current research relevant.

In line 42-44, the sentence “Taste buds are the peripheral organs of gustation and are mainly located in the tongue epithelium; however, they are also present elsewhere in the oral cavity and digestive organs [5]” is flawed. If the authors were talking about distribution of taste buds outside of oral cavity, larynx and esophagus are the appropriate organs to mention. If the authors were talking about distribution of taste cells, multiple organs could be included. Citations need to be double checked and accurate.  

In line 49, “sapid stimuli” is not the right term. Not all gustatory stimuli are “sapid”.

In line 57, “stem cells” is not the appropriate term to describe type IV taste bud cells because these cells do not proliferate and self-renew. They are precursors of type I-III taste cells.

In lines 60-61, is it appropriate to say “warmth or cooling” is elicited by “chemical compounds”?

In lines 80-82, given that “Several reports have indicated that XXXX, and that the oral taste bud cells express cannabinoid receptors [7], [22], [23], [24].”, it is not clear about the novelty and why the authors repeated the study.

It is not clear why the authors use terms “papillae vallatae” and “papillae foliatae” that are different from the ones that are most commonly used in the literature. And the use of authors’ terms is not consistent throughout the manuscript.

3.      Results:

In Figure 1E, multiple bands are seen for TRPA1 making readers wonder the specificity of the antibody.

In lines 141-142, is it appropriate to say “Approximately 25 dapi-labeled nuclei could be counted in single sections of a single vallate papilla;”? Did authors meant to talk about a single taste bud?

In Supplementary figure 1, the data do not support the description in lines 155-156 “SP-positive fibers were seen in contact with taste cells”. First of all, taste cells were not labeled; secondly the “contact” needs a high-resolution image to show it if it is true.

In line 168, was “(b)” meant to be “(B)”?

In line 196, was “€” meant to be “(E)”?

In Figure 6, it is hard for readers to see the images showing TRPV1 and TRPA1-positive cells are in the “most superficial” epithelial cells (line 235).

In Figure 7, it would be helpful to show a low-power image helping readers to understand the relative location of lingual ganglia under the circumvallate papilla.

4.      Discussion:

Overall, it needs to be improved significantly.

There are multiple paragraphs that are composed of a single sentence.

Two subtitles were placed for the later parts. It would be good to have a subtitle for the large part in the beginning.

In lines 257-263, the single long sentence in the whole paragraph is very difficult to understand.

In line 281, “The Authors” could be replaced by “We”.

In lines 390-391, did authors mean to say “these receptors at the gustatory organ level” by “these receptors at a gustatory level”?

5.      Materials and Methods:

In line 399, the values “10,9” and “2,2” are difficult to understand.

In lines 405 and 407, 4C° needs to be corrected.

In line 455, is “CN2R” correct?

In line 482, figure citation “Fig. F” is not clear to readers.

In line 500, did authors mean to say “Tissues at the amount of 50 mg”?

In line 509, the meaning of “RT” is not clear.

6.      Conclusion:

In lines 527-530, the logic, meaning and necessity of this long sentence are not clear.

7.      It would be good to provide a justification for using the pig species and the young piglets.  

Comments on the Quality of English Language

Some terms in taste biology are different from those commonly used in literature.

Spelling errors need to be corrected.

Some long sentences are difficult to understand (meaning and logic).

Author Response

The authors aimed to investigate the expression of cannabinoid receptors and cannabinoid-related receptors in taste buds and other lingual tissues. The results indicate that the immunosignals of these receptors were expressed in the taste bud cells and the surrounding epithelial cells. The extra-papillary epithelium also showed strong immunolabeling for these receptors. The data suggests potential roles of cannabinoid system in taste perception. The writing of the article needs to be improved significantly.

  1. Abstract: Conclusions in line 24-26 are not clear regarding why the “presence of CB-receptors in the extra-gustatory epithelial cells” indicate “their potential role in chemesthesis. Is it the somatosensory nerve fibers or the “extra-gustatory epithelial cells” that transduce the chemesthesis signals?

Response: It appears that keratinocytes/epithelial cells may also play a role in chemesthesis by releasing ATP which stimulates neighboring sensory nerve endings (Sondersorg et al., 2014, The Journal of Biological Chemistry, Vol. 289, No. 25, pp. 17529–17540, June 20, 2014; Viana F., 2011, Chemosensory properties of the trigeminal system, ACS Chem Neurosci, 2(1):38-50, doi: 10.1021/cn100102c).

In our manuscript (lines 66-68), we wrote: 'Given this broad range of sensations, it is not surprising that the receptor mechanisms subserving chemesthesis are equally diverse and are present in various components, including sensory nociceptors, other free nerve endings, and keratinocytes.”

  1. Introduction:

Overall, it could be improved by being more concise and current research relevant.

Response: The introduction has been reduced accordingly.

In line 42-44, the sentence “Taste buds are the peripheral organs of gustation and are mainly located in the tongue epithelium; however, they are also present elsewhere in the oral cavity and digestive organs [5]” is flawed. If the authors were talking about distribution of taste buds outside of oral cavity, larynx and esophagus are the appropriate organs to mention. If the authors were talking about distribution of taste cells, multiple organs could be included. Citations need to be double checked and accurate. 

Response: The Reviewer is right. The sentence has been modified (line 44).

In line 49, “sapid stimuli” is not the right term. Not all gustatory stimuli are “sapid”.

Response: “Sapid stimuli” has been changed to “gustatory stimuli” (line 50).

In line 57, “stem cells” is not the appropriate term to describe type IV taste bud cells because these cells do not proliferate and self-renew. They are precursors of type I-III taste cells.

Response: We thank the Reviewer for this precise clarification. The sentence has been changed accordingly (line 58).

In lines 60-61, is it appropriate to say “warmth or cooling” is elicited by “chemical compounds”?

Response: We think that it is appropriate. There are molecules, such as camphor oil or menthol, which do not heat (or cool) the body but which give, on a certain surface of the body, a sensation of heat or freshness elicited by the activation of specific thermoreceptors.

In lines 80-82, given that “Several reports have indicated that XXXX, and that the oral taste bud cells express cannabinoid receptors [7], [22], [23], [24].”, it is not clear about the novelty and why the authors repeated the study.

Response: With respect, we may not fully understand the Reviewer's observation. The fact that the ECS is expressed in mouse taste buds does not necessarily imply that it is also expressed in other species. It is important to expand the knowledge gained from rodent studies to other animals, such as dogs, horses, pigs, and humans.

For instance, as we mentioned in the text, TRPV1 is not expressed in the taste receptor cells of rodents, whereas it is present in human taste cells from fungiform papillae. Our study is the first to explore the presence of ECS receptors in pigs, as noted on line 95. This represents a pioneering effort which we hope will encourage additional research into developing markers for identifying the various cell types in pig taste buds.

As discussed in the manuscript, understanding the specific distribution of ECS receptors in pigs lays the groundwork for additional studies. These could investigate new nutritional ligands and promote the development of new avenues to enhance animal welfare within animal production."

It is not clear why the authors use terms “papillae vallatae” and “papillae foliatae” that are different from the ones that are most commonly used in the literature.

Response: We work on animals and we are used to utilizing the terminology described in NAV (Nomina Anatomica Veterinaria). We have sometimes used terms utilized in the literature and we have been criticized. Therefore, it is better to apply NAV terminology.

And the use of authors’ terms is not consistent throughout the manuscript.

Response: We tried to verify the consistency of the terms throughout the manuscript by applying the terminology of Nomina Anatomica Veterinaria (2017).

  1. Results:

In Figure 1E, multiple bands are seen for TRPA1 making readers wonder the specificity of the antibody.

Response: We agree with the Reviewer's concern. Nevertheless, we have added the reference molecular weights next to the bands recognized by the antibodies as suggested by Novus NB for the Rabbit 100-91319 antibody. The anti-TRPA1 antibody recognized a major band in the vicinity of 100 kDa which was present in all the pig papilla samples and in the positive controls, such as mouse dorsal root ganglion (DRG) neurons (Fig. 1E). It should be noted that all the figures, negative controls included, are original without having been cropped. We would also like to emphasize that in terms of specificity, all the antibodies we used had been widely tested using western blot by others. See the following references:

  1. A) Adam RJ, Xia Z, Pravoverov K et al. Sympatho-excitation in Response to Cardiac and Pulmonary Afferent Stimulation of TRPA1 Channels is Attenuated in Chronic Heart Failure Rats Am. J. Physiol. Heart Circ. Physiol. 2019-02-01 [PMID: 30707612] (WB, ICC/IF, Rat)
  2. B) Liu Q, Guo S, Huang Y et al. Inhibition of TRPA1 Ameliorates Periodontitis by Reducing Periodontal Ligament Cell Oxidative Stress and Apoptosis via PERK/eIF2 alpha/ATF-4/CHOP Signal Pathway Oxidative medicine and cellular longevity 2022-06-10 [PMID: 35720191] (WB, Human)
  3. C) Duan Z, Zhang J, Li J et al. Inhibition of microRNA-155 Reduces Neuropathic Pain During Chemotherapeutic Bortezomib via Engagement of Neuroinflammation Front Oncol 2020-03-31 [PMID: 32296644] (WB, Rat)

The best control that could be used was an antigen peptide but unfortunately we did not have it and, for this reason, we used a tissue (DRG neurons) in which we knew that TRPA1 was highly expressed.

In lines 141-142, is it appropriate to say “Approximately 25 dapi-labeled nuclei could be counted in single sections of a single vallate papilla;”? Did authors meant to talk about a single taste bud?

Response: We totally agree with the Reviewer. We meant in a single taste bud, and not in a single papilla. The text has been changed accordingly.

In Supplementary figure 1, the data do not support the description in lines 155-156 “SP-positive fibers were seen in contact with taste cells”. First of all, taste cells were not labeled; secondly the “contact” needs a high-resolution image to show it if it is true.

Response: We agree with the Reviewer's concern. The text has been modified to “SP-positive fibers were seen in proximity of unidentified cells of the taste buds”.

In line 168, was “(b)” meant to be “(B)”?

Response: We changed “b” to “B”.

In line 196, was “€” meant to be “(E)”?

Response: We changed “€” to “E”.

In Figure 6, it is hard for readers to see the images showing TRPV1 and TRPA1-positive cells are in the “most superficial” epithelial cells (line 235).

Response: The Reviewer is right because we did not specify that those cryosections were obtained by means of a tangential cut of the tongue (by "tangential" we mean a superficial cut parallel to the surface of the tongue). This specification has been added to the Fig. 6 legend.

In Figure 7, it would be helpful to show a low-power image helping readers to understand the relative location of lingual ganglia under the circumvallate papilla.

Response: We prepared a new Supplementary figure (Supplementary Figure S3) in which the neuronal ganglia are visible close to the papilla vallata.

  1. Discussion:

Overall, it needs to be improved significantly.

Response: We are sorry that the Reviewer did not like our article in general. We deleted some sentences which were not sufficiently specific.

There are multiple paragraphs that are composed of a single sentence.

Response: We revised the structure of the text, and improved it.

Two subtitles were placed for the later parts. It would be good to have a subtitle for the large part in the beginning.

Response: We agree with the Reviewer; a new subtitle (3.1) has been added: 3.1. Cannabinoid and cannabinoid-related receptors in gustatory epithelial cells

In lines 257-263, the single long sentence in the whole paragraph is very difficult to understand.

Response: We thank the Reviewer. The paragraph has been improved it.

In line 281, “The Authors” could be replaced by “We”.

Response: There is a misunderstanding because it is the Authors of the above-mentioned study who speculate that… etc etc.  However, this sentence has been deleted from the manuscript.

In lines 390-391, did authors mean to say “these receptors at the gustatory organ level” by “these receptors at a gustatory level”?

Response: We thank the Reviewer for this correction. The text has been modified accordingly.

  1. Materials and Methods:

In line 399, the values “10,9” and “2,2” are difficult to understand.

Response: The “commas” have been changed to “decimal points”.

In lines 405 and 407, 4C° needs to be corrected.

Response: The text has been modified accordingly.

In line 455, is “CN2R” correct?

Response: In the web site of Thermo Fisher there is CN2R; we changed CN2R to CB2R.

In line 482, figure citation “Fig. F” is not clear to readers.

Response: “Fig. F” had been changed to “Fig. 1F”.

In line 500, did authors mean to say “Tissues at the amount of 50 mg”?

Response: We couoldn’t find this in the text but we’d say that it should be “Tissues in the amount of 50 mg”.

In line 509, the meaning of “RT” is not clear.

Response: The acronym (RT, room temperature) was explained on line 426.

  1. Conclusion:

In lines 527-530, the logic, meaning and necessity of this long sentence are not clear.

Response: The sentence has been deleted.

  1. It would be good to provide a justification for using the pig species and the young piglets.

Response: Working with tissues from young animals is preferred to avoid the deposition of nonspecific autofluorescent granules within the cells (such as lipofuscin), which could interfere with immunolabeling, especially when carrying out immunofluorescence. Moreover, as mentioned in the text, maternal milk contains high levels of endocannabinoids; therefore, the ECS and cannabinoids may play a role in early-life feeding in mammals.

English: Some terms in taste biology are different from those commonly used in literature.

Spelling errors need to be corrected. Some long sentences are difficult to understand (meaning and logic).

Response: Since we are not native speakers, we always use an expert English reviewer. We have provided Molecules with a linguistic review certificate. It is the same reviewer (expert in medical-scientific subjects) who has worked with us in preparing many articles which have never been contested for their English. I would like to point out that Molecules edited our Word file, also somewhat changing the English which may have created some inconvenience for the Reviewer; in fact, we noticed that often some points were missing and that the sentences had been rearranged differently from what the English reviewer had written.

Round 2

Reviewer 1 Report

Comments and Suggestions for Authors

The authors have addressed all my concerns.

Comments on the Quality of English Language

English language is fine.

Author Response

Response: We thank the Reviewer for appreciating and accepting our changes to the text.

Reviewer 2 Report

Comments and Suggestions for Authors

After addressing many of the concerns, the clarity of the article has been improved. However, the discussion still contains single-sentence paragraphs that sound like notes for the writers themselves. The texts need to be structured for better logic and connection.

Author Response

Response: We thank the Reviewer for the comments. Modifications were made accordingly to improve the structure and logic of the text.